# Identification of Genomic Safe Harbors in the Anhydrobiotic Cell Line, Pv11

**DOI:** 10.3390/genes13030406

**Published:** 2022-02-24

**Authors:** Yugo Miyata, Shoko Tokumoto, Tomohiko Arai, Nurislam Shaikhutdinov, Ruslan Deviatiiarov, Hiroto Fuse, Natalia Gogoleva, Sofya Garushyants, Alexander Cherkasov, Alina Ryabova, Guzel Gazizova, Richard Cornette, Elena Shagimardanova, Oleg Gusev, Takahiro Kikawada

**Affiliations:** 1Division of Biomaterial Sciences, Institute of Agrobiological Sciences, National Agriculture and Food Research Organization (NARO), Tsukuba 305-0851, Japan; miyata.mche@tmd.ac.jp (Y.M.); tokumotos023@affrc.go.jp (S.T.); cornette@affrc.go.jp (R.C.); 2Department of Medical Chemistry, Medical Research Institute, Tokyo Medical and Dental University, Tokyo 113-8510, Japan; 3Department of Integrated Biosciences, Graduate School of Frontier Sciences, The University of Tokyo, Kashiwa 277-8562, Japan; 9047097639@edu.k.u-tokyo.ac.jp (T.A.); 4182125849@edu.k.u-tokyo.ac.jp (H.F.); 4Center of Life Sciences, Skolkovo Institute of Science and Technology, 121205 Moscow, Russia; nurislam.shaikhutdinov@skoltech.ru (N.S.); garushyants@gmail.com (S.G.); cherkasovav@yandex.ru (A.C.); 5Regulatory Genomics Research Center, Institute of Fundamental Medicine and Biology, Kazan Federal University, 420012 Kazan, Russia; ruselusalbus@gmail.com (R.D.); negogoleva@gmail.com (N.G.); urban-nomad@yandex.ru (A.R.); grgazizova@gmail.com (G.G.); ryukula@gmail.com (E.S.); gaijin.ru@gmail.com (O.G.); 6Endocrinology Research Center, 115478 Moscow, Russia; 7Graduate School of Medicine, Juntendo University, Tokyo 113-8421, Japan; 8Laboratory for Transcriptome Technology, RIKEN Center for Integrative Medical Sciences, RIKEN, Yokohama 230-0045, Japan

**Keywords:** Pv11 cells, genomic safe harbor sites, anhydrobiosis, transgenesis, cell engineering

## Abstract

Genomic safe harbors (GSHs) provide ideal integration sites for generating transgenic organisms and cells and can be of great benefit in advancing the basic and applied biology of a particular species. Here we report the identification of GSHs in a dry-preservable insect cell line, Pv11, which derives from the sleeping chironomid, *Polypedilum vanderplanki*, and similar to the larvae of its progenitor species exhibits extreme desiccation tolerance. To identify GSHs, we carried out genome analysis of transgenic cell lines established by random integration of exogenous genes and found four candidate loci. Targeted knock-in was performed into these sites and the phenotypes of the resulting transgenic cell lines were examined. Precise integration was achieved for three candidate GSHs, and in all three cases integration did not alter the anhydrobiotic ability or the proliferation rate of the cell lines. We therefore suggest these genomic loci represent GSHs in Pv11 cells. Indeed, we successfully constructed a knock-in system and introduced an expression unit into one of these GSHs. We therefore identified several GSHs in Pv11 cells and developed a new technique for producing transgenic Pv11 cells without affecting the phenotype.

## 1. Introduction

Transgene integration, one of the most commonly used and effective techniques in biological research, can be achieved by either of two approaches: random integration of the gene of interest or site-specific knock-in. Random integration is the simpler method, but it can have unwanted and unpredictable effects on the host cell phenotype, depending on the integration site. In contrast, site-specific knock-in results in more controlled outcomes and has been facilitated by recent advances in genome-editing technologies. Furthermore, in some species and cell lines, several research groups have developed genomic sites called genomic safe harbors (GSHs) that enable transgenes to function as designed without adverse effects on the host [1,2,3,4,5,6]. Transgene integration into such GSHs has promoted fundamental and applied biological research, but identifying GSHs is still challenging, especially in non-model species, owing to insufficient development of the genome database and genetic engineering tools.

GSHs are sites in the genome where new genetic material can be integrated without affecting the host phenotype and where the newly integrated material functions as designed [1]. For example, in basic research, a variety of genetic materials have been integrated into a mouse GSH, the Rosa26 locus, allowing exogenous gene expression or highly targeted gene knockdown experiments [7]. In applied research, GSHs have been used to establish cells that can stably produce proteins of medical interest, for example, therapeutic antibodies [4,8] and species-specific glycosylated proteins [9]. In addition, some researchers have proposed a therapy for genetic diseases using cells that are genetically modified via human GSHs [10]. Therefore, the identification and exploitation of GSHs can significantly advance our understanding of the biology of a particular species as well as leading to the development of new biotechnological applications.

GSHs have usually been identified by a four-step approach [4,11,12]: (i) creation of a transgenic cell pool by random integration; (ii) selection of cells showing stable transgene expression and no/few phenotype changes due to the exogenous gene material; (iii) genome analysis of the selected cells; (iv) confirmation of whether the integration sites are GSHs by a site-specific knock-in method. However, there are a number of difficulties in executing this strategy. Thus, for step (i), suitable genetic engineering tools are needed to create transfectants, and these might not be available for the species of interest. Step (ii) is time-consuming and requires a major effort to monitor the functional and phenotypic stability of the integrated genetic material and the transformant cells, respectively. For step (iii), a genome sequence and database are normally required to identify integration sites, while for step (iv), an appropriate site-specific knock-in method, such as the CRISPR/Cas9 system, must be available for the subject species. Therefore, GSH identification is not a straightforward process, especially in non-model species.

Pv11 is a culturable cell line derived from an insect, the sleeping chironomid *P. vanderplanki*, which inhabits semi-arid regions in Africa [13]. *P. vanderplanki* larvae display extreme desiccation tolerance [14], and Pv11 has inherited this ability, such that the cells can be preserved in the dry state at room temperature, while retaining their ability to proliferate once rehydrated [13]. When Pv11 cells are dried, any exogenous protein they contain can be preserved at room temperature for up to 372 days; thus, because of their desiccation tolerance, Pv11 cells potentially have a number of industrial applications, for instance, as water-free storage containers for biomaterials at room temperature [15]. The identification of GSHs in Pv11 cells will provide a major step towards such applications.

To explore the molecular mechanisms underlying the desiccation tolerance of Pv11 cells, we have exploited several gene-manipulation techniques, including a random integration method [16] and a site-specific knock-in method using the CRISPR/Cas9 system [17,18], and have generated several key resources. We have produced a transgenic cell pool, so-called KH cells, by random integration of an AcGFP1 (*Aequorea coerulescens* GFP)-expressing plasmid [16]. Furthermore, we have a well-annotated genomic database and have developed the CRISPR/Cas9 system in Pv11 cells [19]. Thus, the materials needed to identify GSHs of Pv11 cells are already in place.

Here we describe the successful identification of GSHs in Pv11 cells. Following the approach used by others, we derived subpopulations from a KH cell pool and then analyzed the transgene integration sites in these cells at the whole-genome level. We found four candidate GSHs in Pv11 cells. We confirmed their potential using an AcGFP1 ex-pression unit consisting of a functional promoter, the AcGFP1-coding sequence, and the polyadenylation signal, which were inserted into each site using the CRISPR-mediated targeted knock-in method. The proliferation rate and desiccation tolerance of the knock-in cells were analyzed, leading to the identification of three GSHs on chromosome 1 (Chr1) of the Pv11 genome, Chr1:21143572, Chr1:21155382, and Chr1:21164645. Knock-in cells containing exogenous material at the Chr1:21164645 site displayed stable expression of the transgene after more than one year in culture. In addition, we constructed a basic donor vector system to knock-in exogenous genetic material into the Chr1:21164645 site. These results demonstrate the construction of a new gene-manipulation system for Pv11 cells, which should facilitate future advances in the basic biology and applied biotechnology of these cells.

## 2. Materials and Methods

### 2.1. Cell Culture

Pv11 and Pv11-KH cells were grown as described previously [18]. Briefly, for their culture, IPL-41 medium (Thermo Fisher Scientific, Waltham, MA, USA) supplemented with 2.6 g/L tryptose phosphate broth (Becton, Dickinson and Company, Franklin Lakes, NJ, USA), 10% (*v/v*) fetal bovine serum, and 0.05% (*v/v*) of an antibiotic and antimycotic mixture (penicillin, amphotericin B, and streptomycin; MilliporeSigma, Burlington, MA, USA) was used, and the medium is designated hereafter as complete IPL-41 medium. A density of 3 × 10^5^ cells/mL were seeded into a cell culture flask with plug seal cap and grown at 25 °C for 6–7 days.

### 2.2. Cloning a Subpopulation of KH Cells

Single cell sorting was performed using a MoFlo Astrios cell-sorter (Beckman Coulter Life Sciences, Indianapolis, IN, USA) as described previously [18]. Briefly, the cells were stained with DAPI (Dojindo, Kumamoto, Japan), and DAPI and AcGFP1 were excited with 355 and 488 nm lasers, respectively. In total, 1000 wild-type Pv11 cells were seeded as a feeder layer in each well of a 96-well plate prior to sorting. The sorted and the feeder cells were grown for two weeks, and then a second sorting (bulk cell sorting) was performed to eliminate the feeder cells.

### 2.3. Desiccation and Rehydration

Pv11 cells were subjected to desiccation-–rehydration as described previously [13]. Briefly, cells were incubated in preconditioning medium (600 mM trehalose containing 10% (*v*/*v*) complete IPL-41 medium) for 48 h at 25 °C. Forty-microliter aliquots of the cell suspension were dropped into 35-mm petri dishes, and the dishes were desiccated and maintained at <10% relative humidity and 25 °C for more than seven days. An hour after rehydration by complete IPL-41 medium, cells were stained with propidium iodide (PI; Dojindo) and Hoechst 33342 (Dojindo), and images were acquired using a conventional fluorescence microscope (BZ-X700; Keyence, Osaka, Japan). The survival rate was calculated as the ratio of the number of live cells (Hoechst-positive and PI-negative) to that of total cells (Hoechst-positive).

### 2.4. Proliferation Analysis

Pv11 cells and the other cell lines were seeded at a density of 1 × 10^5^ cells/mL, and the live cell numbers were counted after staining with PI and Hoechst 33342 (Dojindo), as described in Section 2.3.

### 2.5. High-Molecular-Weight DNA Extraction and Purification

High-molecular-weight DNA was extracted with NucleoBond HMW DNA Kit (TaKaRa Bio, Shiga, Japan). The DNA solution was further treated with a Short Read Eliminator Kit (Circulomics, Baltimore, MD, USA) to deplete short DNA fragments. DNA concentrations were measured using a Qubit2.0 Fluorometer (Thermo Fisher Scientific, Waltham, MA, USA).

### 2.6. Library Preparation for MinION Sequencing

Library preparation was performed using a Ligation Sequencing Kit (version SQK-LSK109; Oxford Nanopore, Oxford, UK), following the manufacturer’s instructions. These libraries were sequenced on the MinION platform, using a Flow Cell R9.4.1 and MinKNOW software (20.06.5).

### 2.7. Base-Calling and Data Analysis

The raw data were acquired as fast5 files, and base-called with Guppy basecaller software (v4.2.2). Low quality and short reads (Phred score < 7 and length < 1000 bp, respectively) were removed using NanoFilt (version 2.7.1). To detect the integration sites of a transfected plasmid (Appendix A) [16], we performed the following three steps: (i) to extract the reads containing a sequence/sequences of the exogenous DNA material, a BLASTn (2.10.1+) search was carried out, using the exogenous sequence as the query against the database of sequencing reads that passed the quality filters (E-value < 1 × 10^−30^, -outfmt 6); (ii) 500 bases of sequence upstream and downstream of the integrated exogenous sequences found in step (i) reads were extracted using SeqKit (version 0.13.2) with the following parameters: grep -nrp, subseq -r; (iii) the integration sites in the genome were found by BLASTn search of these upstream and downstream regions (see step (ii)) against the Pv11 genome (Genbank Assembly Accession ID: GCA_018290105.1, E-value < 1 × 10^−30^, -outfmt 6, max_target_seq 1, max_hsps 1). The results were visualized using the DensityMap tool [20].

### 2.8. Expression Vectors

For Cas9 expression, the previously constructed vector, pPv121-hSpCas9, was used; gRNA expression vectors were also constructed as reported previously [17]. Briefly, pPvU6b-DmtRNA-BbsI was digested with BbsI, and annealed oligonucleotides were ligated into the cut vector. Vector names and oligonucleotides used for the gRNA expression vector construct are listed in Appendix A.

### 2.9. Genomic PCR and Sanger Sequencing Analysis

Accurate genome sequences around GSH candidates were analyzed by genomic PCR and Sanger sequencing. Pv11 cell genomic DNA was extracted with a NucleoSpin Tissue kit (Takara Bio) and subjected to PCR using specific primer sets (Appendix A). After gel purification of the PCR products, sub-cloning was carried out using a TOPO cloning kit (Thermo Fisher Scientific, Waltham, MA, USA), and the plasmids were sequenced.

### 2.10. Donor Vector Construction

Donor vectors containing expression units for AcGFP1 or zeocin resistance (ZeoR) were constructed using PCR, a HiFi Assembly kit (New England BioLabs, Ipswich, MA, USA) and a Zero Blunt TOPO PCR cloning kit (Thermo Fisher Scientific, Waltham, MA, USA). As a PCR template, the AcGFP1 or ZeoR expression unit of a previously constructed vector, pPv121-AcGFP1-Pv121-ZeoR, was used [21]. Primers in the PCR included the gRNA-target and 40 bp homology sequences of each knock-in site. PCR products were cloned into pCR4 Blunt-TOPO (Thermo Fisher Scientific, Waltham, MA, USA); all primer sequences and vector names are listed in Appendix A.

Targeted integration at the Chr1:21164645#9 site was performed using donor vectors containing the bidirectional AcGFP1 and HaloTag expression unit and different homology arms (all PCR primers are listed in Appendix A). Donor vectors were constructed using PCR, a HiFi Assembly kit (New England BioLabs) and a DNA Ligation Kit Mighty Mix (Takara Bio), based on pCR4-Pv.00443#1-P2A-GCaMP3 and pCRII-Pv.00443#1-P2A-AcGFP1-P2A-ZeoR [18]. The former vector was digested with SpeI and NotI. The 3910 bp backbone vector and the following two PCR fragments were assembled: (i) the 1000-base left homology arm with the gRNA-target sequence and SpeI site at the 5′- and 3′-end, respectively; (ii) the 1000-base right homology arm with the NotI site and the gRNA-target sequence at the 5′- and 3′-end, respectively. In the HiFi assembly, the original SpeI and NotI sites of pCR4-Pv.00443#1-P2A-GCaMP3 were eliminated. The assembled vector was named pCR4-21164645#9_1kbpHA-SpeI_NotI, and further used as a PCR template to make PCR fragments with the 40–750 base homology arms. The PCR products were inserted into the digested 3910 bp backbone vector (pCR4-21164645#9_40bpHA-SpeI_NotI, pCR4-21164645#9_125bpHA-SpeI_NotI, pCR4-21164645#9_250bpHA-SpeI_NotI, pCR4-21164645#9_500bpHA-SpeI_NotI, pCR4-21164645#9_750bpHA-SpeI_NotI). In the case of the vector without a homology arm, AcGFP1 was used as a PCR template, which provided a sufficient length of fragment for HiFi assembly in the next step. A SpeI site and the gRNA-target sequence were added at the 5′-end of the fragment, while a NotI site and the gRNA-target sequence were added at the 3′-end. The PCR product was inserted into the digested 3910 bp backbone vector (pCR4-21164645#9_0bpHA-SpeI_AcGFP1_NotI).

Next, pCRII-Pv.00443#1-P2A-AcGFP1-P2A-ZeoR was digested with HindIII and XbaI. The 3407 bp backbone vector and the following two PCR fragments were assembled: (i) the 121 promoter-controlled HaloTag expression unit with a SpeI site at the 5′-end, (ii) the 121 promoter-controlled AcGFP1 expression unit with a NotI site at the 3′-end. The assembled vector was named pCRII-SpeI-HaloTag-121-121-AcGFP1-NotI.

Lastly, pCR4-21164645#9_0bpHA-SpeI_AcGFP1_NotI, pCR4-21164645#9_40bpHA-SpeI_NotI, pCR4-21164645#9_125bpHA-SpeI_NotI, pCR4-21164645#9_250bpHA-SpeI_NotI, pCR4-21164645#9_500bpHA-SpeI_NotI, pCR4-21164645#9_750bpHA-SpeI_NotI, and pCRII-SpeI-HaloTag-121-121-AcGFP1-NotI were digested with SpeI and NotI and ligated. The vectors were named pCR4-21164645#9_0bpHA-HaloTag-121-121-AcGFP1, pCR4-21164645#9_40bpHA-HaloTag-121-121-AcGFP1, pCR4-21164645#9_125bpHA-HaloTag-121-121-AcGFP1, pCR4-21164645#9_250bpHA-HaloTag-121-121-AcGFP1, pCR4-21164645#9_500bpHA-HaloTag-121-121-AcGFP1, pCR4-21164645#9_750bpHA-HaloTag-121-121-AcGFP1, pCR4-21164645#9_1kbpHA-HaloTag-121-121-AcGFP1 (the complete sequences are shown in Appendix A).

### 2.11. Transfection and Site-Specific Transgene Knock-In and Establishment of Clonal Cell Lines

Transfection for site-specific knock-in was carried out using a NEPA21 Super Electroporator (Nepa Gene, Chiba, Japan) as described previously [21]. In total, 5 μg each of the gRNA- and SpCas9-expression vectors plus 0.03 pmol each of donor vectors were transfected into Pv11 cells. Five days after transfection, the cells at a density of 1 × 10^5^ cells per mL were treated with 400 μg/mL zeocin. After the zeocin selection, the medium was changed to complete IPL-41 medium, and the cells were incubated for an additional two weeks. To establish clonal cell lines, single cell sorting was performed, using a MoFlo Astrios cell-sorter (Beckman Coulter Life Sciences), as described in Section 2.2. The sorted and the feeder cells were grown in the complete IPL-41 medium without zeocin for two weeks and then treated with zeocin for two weeks to eliminate the feeder cells. Once the establishment of clonal cell lines was confirmed, they were cultured in the complete IPL-41 medium without zeocin. A portion of each cell line was cryopreserved with CELLBANKER 1plus (Takara Bio). The remaining cells were passaged over one year in the complete IPL-41 medium without zeocin. Hence, cell lines cultured continuously for more than one year, the above-mentioned cryopreserved cells thawed and cultured for only three weeks, and intact Pv11 cells were used to analyze the expression level of AcGFP1 protein by the cell-sorter and cell phenotypes of survival rate and proliferation ability.

### 2.12. Optimization of the Homology Arm Length for Maximum Knock-In Efficiency

For knock-in of both HaloTag and AcGFP1 expression units at Chr1:21164645, 0.03 pmol of a donor vector containing the expression units were transfected with 5 μg each of the gRNA- and SpCas9-expression vectors plus 0.03 pmol of a donor vector containing the ZeoR expression unit with 40 bp homology arms. Five days after transfection, the cells at a density of 1 × 10^5^ cells per mL were treated with 400 μg/mL zeocin. After 10 days in culture with zeocin selection, the cells were subjected to flow cytometry analysis by a CytoFLEX S (Beckman Coulter Life Sciences). For HaloTag labeling, HaloTag TMRDirect Ligand (Promega, Fitchburg, WI, USA) was added to the medium 16 h before the analysis. The fluorescence of DAPI, AcGFP1, and HaloTag TMRDirect Ligand was detected by excitation with a 405, 488, and 561 nm laser, respectively.

### 2.13. Statistical Analysis

All data were expressed as mean ± standard deviation (SD). Differences between two groups were examined for statistical significance using Student’s *t*-test. Statistical significance among more than three groups was examined by ANOVA followed by a Tukey post-hoc test. A *p*-value < 0.05 denoted a statistically significant difference. GraphPad Prism 8 software (GraphPad, San Diego, CA, USA) was used for the statistical analyses.

## 3. Results

### 3.1. Cloning Subpopulations with Improved Anhydrobiotic Ability from a KH Cell Pool

To isolate clonal KH cell subpopulations, single-cell sorting was performed (Figure 1A). We acquired two cell lines, B2 and 4C, whose survival rates after rehydration were higher than the original KH cells (Figure 1B). The proliferation rates of B2 and 4C were the same and faster than that of KH cells, respectively, although all KH-derived cells grew more slowly than wild-type Pv11 cells (Figure 1C). Thus, the two cell lines displayed a less-impaired phenotype than the original KH cells.

### 3.2. Genome-Wide Analysis and Identification of the Integration Sites in B2 and 4C Lines

Next, to identify the integration sites in the two cell lines, high-molecular-weight genomic DNA was extracted (Appendix A). DNA libraries were prepared and sequenced with a MinION sequencer (Appendix A). As illustrated in Figure 2A, fragmented plasmid sequences were detected throughout the whole genome (Figure 2B, and Appendix A). In contrast, the AcGFP1 expression unit was detected only on chromosome 1 (Chr1; Figure 2B,C, and Appendix A) in both clones. Three of these integration sites, Chr1:280397, Chr1:21155382 and Chr1:21164645, were located in intergenic regions, while a fourth, Chr1:21143572, was located in the intron of transcription unit, *g12121*, whose expression is relatively low in Pv11 cells (Figure 2C, and Appendix A, accession number GSE171333 [17]).

### 3.3. Identification of Genomic Safe Harbors in Pv11

Next, we examined whether genomic integration at the above sites affects the anhydrobiotic ability or proliferation rate of the corresponding cells. As shown in Figure 3A, AcGFP1 and ZeoR expression units were inserted individually into each genomic site in wild-type Pv11 cells by the CRIS-PITCh method [17,18,22]. Although precise integration was not achieved at Chr1:280397 (Appendix A), the other three sites allowed exogenous DNA integration as designed, and this was confirmed by Sanger sequencing (Appendix A). The knock-in cell lines displayed similar desiccation survival rates and proliferation rates to wild-type Pv11 cells (Figure 3B,C). Therefore, the three sites, Chr1:21143572, Chr1:21155382, and Chr1:21164645, were identified as potential GSHs in Pv11.

We then checked the stability of the protein expression level and the cellular phenotypes in a knock-in cell line with a copy of AcGFP1 at Chr1:21164645. The cells were grown for more than one year and, as shown in Figure 4, long-term culture had no effect on AcGFP1 fluorescence intensity (Figure 4A), nor the anhydrobiotic ability (Figure 4B) and proliferation rate (Figure 4C) of the cells.

### 3.4. Construction of a GOI Knock-In System for Pv11

Next, we attempted to construct a gene-of-interest (GOI) expression system at the Chr1:21164645 site. A donor vector containing bidirectional HaloTag and AcGFP1 expression unit was designed, the former as an example of a GOI, and the latter as a marker of successful integration (Figure 5A). In our first attempt, a 40 bp homology arm length was used as shown in Figure 3 and described in previous studies [17,18,22]. However, the integration efficiency was low (7.1 ± 1.3% of the target cells were HaloTag^+^/AcGFP1^+^ cells; Figure 5B,C), possibly because the insert size was much longer than the homology arm length. Therefore, to determine the optimal homology arm length for this knock-in system, we constructed a series of donor vectors with bidirectional HaloTag and AcGFP1 expression unit flanked by homology arms of various lengths (from 0 to 1000 bp, Appendix A; Figure 5A). Each of these donor vectors was transfected with the previously used construct containing the ZeoR expression unit (Figure 3), and after zeocin selection the protein expression levels of HaloTag and AcGFP1 were analyzed (Figure 5B). There was no significant difference in the knock-in efficiencies of the 250, 500, 750, and 1000 bp HA groups (23.7 ± 2.7%, 26.0 ± 2.6%, 26.0 ± 4.3%, and 25.6 ± 2.9% of cells were HaloTag^+^/AcGFP1^+^, respectively), but the 0, 40, and 125 bp HA groups all gave a lower knock-in efficiency (8.8 ± 2.4%, 7.1 ± 1.3%, and 20.7 ± 1.7% of cells were HaloTag^+^/AcGFP1^+^, respectively; Figure 5C). Thus, a homology arm length greater than 250 bp is needed for efficient knock-in of the GOI expression construct at the Chr1:21164645 site.

To check whether the GOI knock-in construct affects anhydrobiosis and cell proliferation, a clonal HaloTag- and AcGFP1-positive cell line was established, and the genotype and phenotype were analyzed. Sanger sequencing of the genome showed precise integration of the donor vector as designed in Figure 5A (Appendix A), while the cells exhibited the same anhydrobiotic ability and proliferation rate as wild-type Pv11 cells (Appendix A).

## 4. Discussion

The major achievements of the current study are: (i) identification of GSHs in Pv11 cells and (ii) the construction of a basic knock-in system at one of the GSHs, Chr1:21164645. To identify GSHs in Pv11 cells, a transgenic cell pool obtained by random integration was used as source material, and two clonal cell lines with less-impaired phenotypic differences than the overall cell population were established. Whole-genome sequencing was performed on the cell lines, identifying four potential GSHs that were then used for knock-in experiments. At three of these sites, precise Cas9-mediated integration was achieved, and all three sites fulfilled the GSH criteria, i.e., there were no deleterious effects on anhydrobiotic ability or proliferation rate following knock-in. Furthermore, one of the GSHs, Chr1:21164645, maintained the function of the integrated gene after more than one year in culture and this site was subsequently used to construct a GOI-expression system. The resources generated in this study should be useful for advancing our understanding of anhydrobiosis, but also for biotechnological applications of Pv11 cells.

We previously reported a method to stably express GOIs in Pv11 cells [17,18]. The method exploits the high-expression system of the endogenous *Pv.00443* gene (as known as *g7775*): GOI coding sequences plus P2A are knocked into the 5′-flanking site of the stop codon of *Pv.00443* resulting in polycistronic expression. Although this approach is appropriate for constitutive expression of GOIs, it cannot be applied to integration of artificial expression units, for example, inducible gene expression systems [23], such as the Tet-On inducible expression system we previously developed in Pv11 cells [24]. Knocking in the system will be invaluable for tight control of the expression of proteins that inhibit the growth rate in Pv11 cells, and thus will facilitate the use of Pv11 cells as water-free storage containers for proteins. Another example is small RNA-expression systems including RNA Polymerase III promoters [25]. Polymerase III promoters have often been used to express shRNAs and gRNAs and can also be used for genome-wide screening [26,27]. Thus, our finding of GSHs in Pv11 cells will be of great benefit in further revealing the molecular mechanisms underlying anhydrobiosis.

We constructed a basic vector to knock-in a GOI at the Chr1:21164645 site (Figure 4) and found that homology arm lengths of 250–500 bp were sufficient for maximum knock-in efficiency. However, this efficiency was still only around 25%. To increase the efficiency, further modification of the knock-in system might be attempted, for example, by inhibition of some protein activities associated with the specific repair of DNA double-strand breaks [28]. Indeed, a recent study reported a drastic shift from non-homologous end joining-/microhomology-mediated end joining-mediated repair processes to a homology-directed repair-mediated process by inhibition of DNA polymerase θ and DNA-dependent protein kinase, which led to a huge improvement in knock-in efficiency [29]. Such methods might also enable simultaneous insertion of exogenous DNAs into multiple GSH sites, or multi-copy integration at the same location, which can lead to increased recombinant protein production in a cell. Thus, improving the efficiency of our knock-in system may be a key factor in biotechnological applications of Pv11 cells.

Exogenous gene integration into genomic sites can occasionally cause changes in genomic structure or gene expression pattern, which in turn can cause alteration in cellular phenotypes (e.g., oncogenesis) [1]. Such phenotypic changes are a major concern, especially in human gene therapy applications [1]. In general, global gene expression patterns are assessed to confirm that the likelihood of any potential phenotypic changes in the future is low [3,10]. In other words, the most important point is whether the intrinsic cellular phenotypes can be maintained after transgenesis. Although we did not examine the gene expression patterns of the transgenic cell lines, intrinsic characteristics of Pv11 cells, namely the ability to anhydrobiosis and proliferation, were maintained even after one year in the cell lines (Figure 3 and Figure 4). Hence, we believe that the current results are sufficient to derive our conclusion.

To begin to understand why the three sites, Chr1:21143572, Chr1:21155382, and Chr1:21164645, behave as GSHs in Pv11 cells, we acquired ATAC-seq data (GEO ID: GSE190481) to assess chromatin accessibility. Appendix A shows that the integration sites of the fragmented plasmid are in or near ATAC-seq peaks, which suggests that random integration of exogenous DNAs is most likely to occur in open chromatin regions. However, we cannot otherwise find any specific features of the GSH regions, such as larger or more consecutive ATAC-seq peaks, that might provide further clues. Further examination of chromatin conformation analysis may be needed, for example, using 5C and Hi-C methods [30,31]. These techniques detect active or inactive chromatin compartments on a genome-wide scale, and the information would help settle the question of why GSHs in Pv11 cells locate in a specific region of Chr1.

## 5. Conclusions

We identified GSHs in Pv11 cells that, when used for exogenous gene integration, had no effect on two phenotypes tested: anhydrobiotic ability and proliferation rate. Testing of cells obtained using one of the GSHs showed that the integrated gene functioned as designed. Furthermore, we constructed a basic tool kit to knock-in a GOI expression unit at this GSH. These resources will facilitate further genome engineering strategies, such as an inducible-expression system, and contribute to advancements in basic biology and applied biotechnology in Pv11 cells.

## Figures and Tables

**Figure 1 genes-13-00406-f001:**
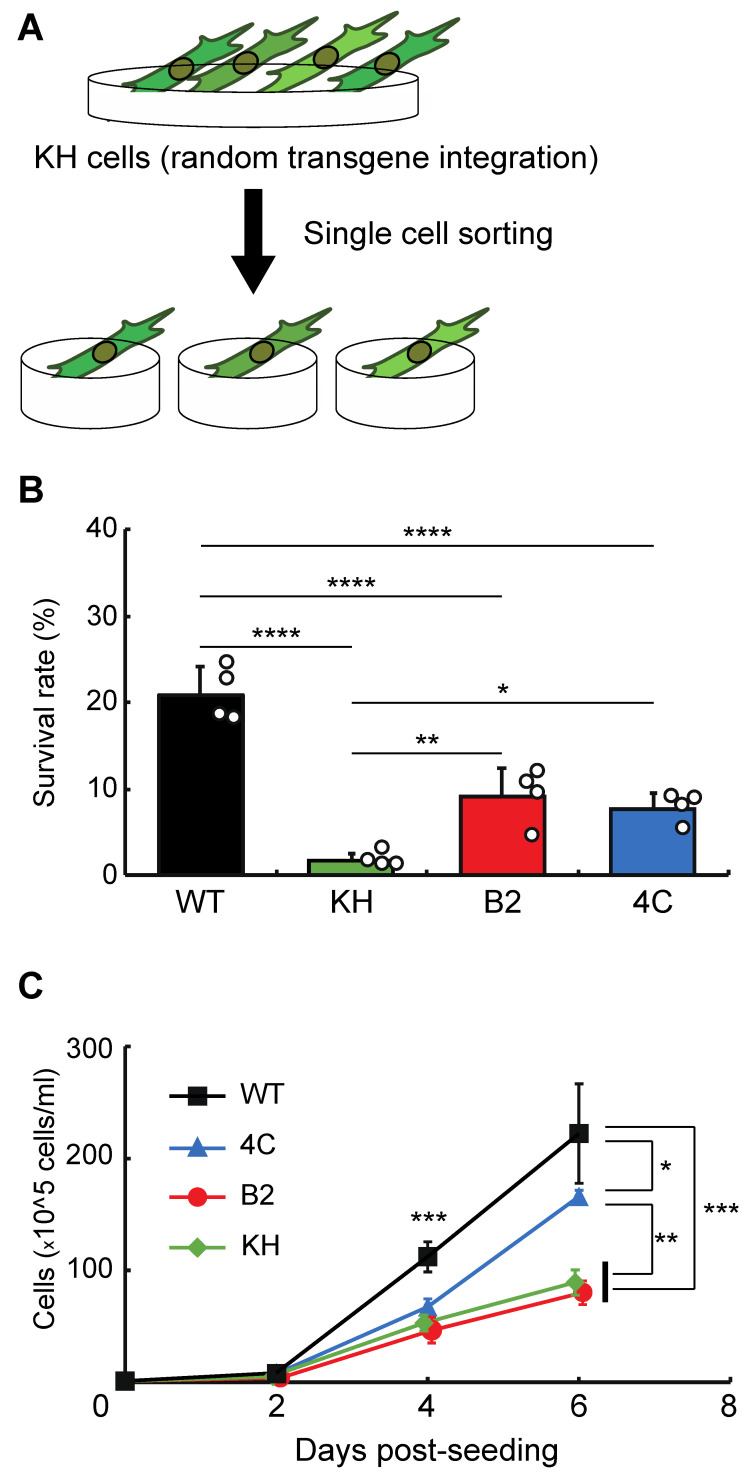
Selection of clonal cell lines from the KH cell pool. (**A**) The experimental scheme is shown. To establish clonal cell lines from KH cells, single-cell sorting was performed. (**B**) The survival rates after desiccation–rehydration treatment of the clonal cell lines, B2 and 4C, are shown. (**C**) The proliferation rates of the B2 and 4C cell lines are shown. Values are expressed as mean ± standard deviation (SD); n = 4 in each group. **** *p* < 0.0001; *** *p* < 0.001; ** *p* < 0.01; * *p* < 0.05.

**Figure 2 genes-13-00406-f002:**
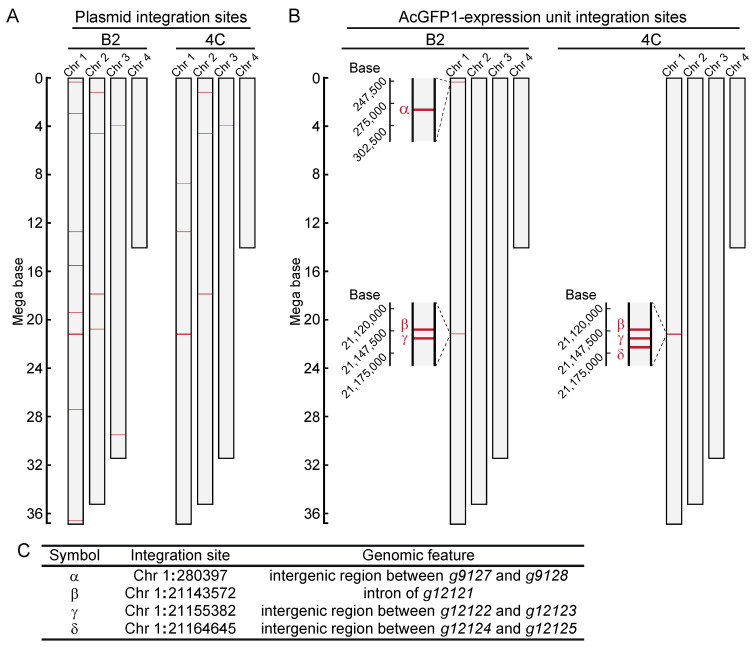
Genome-wide analysis of integration sites of the exogenous plasmid sequence in clonal cell lines, B2 and 4C. (**A**) Integration sites of the plasmid sequence are shown. (**B**) Integration sites of the AcGFP1-expression unit are shown. (**C**) Identified GSH candidates are listed, and their genomic features are described.

**Figure 3 genes-13-00406-f003:**
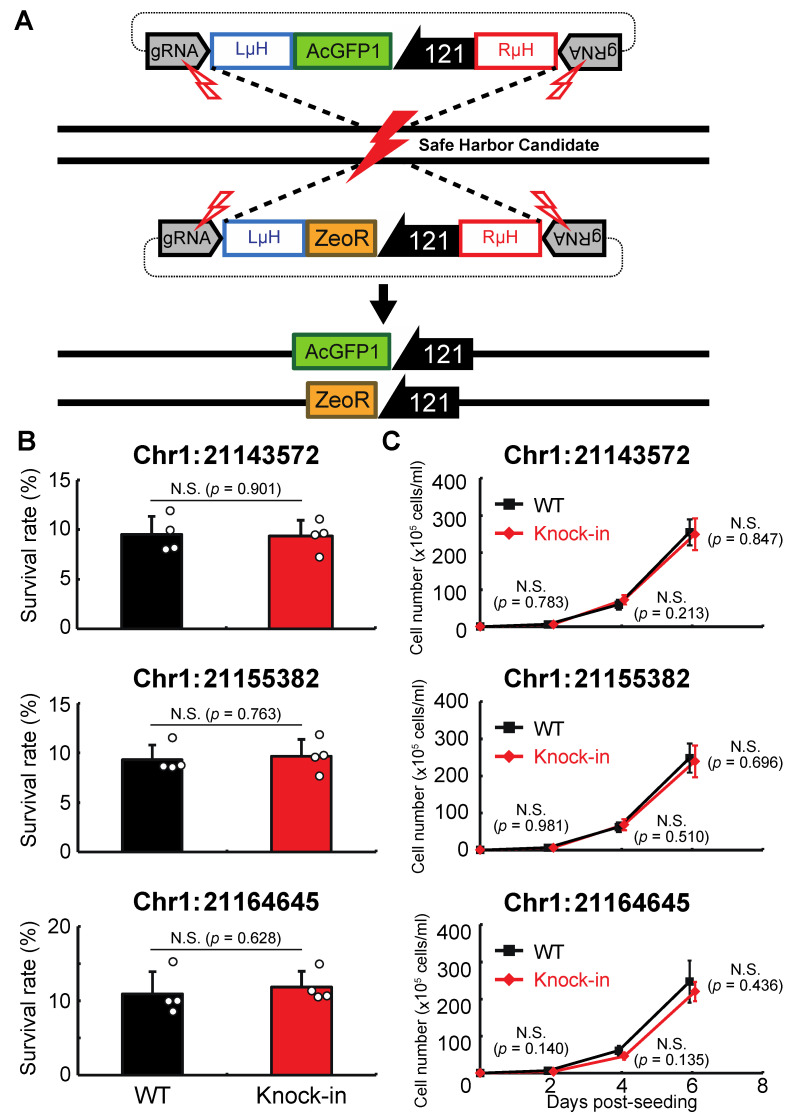
Integration of the expression units of AcGFP1 and zeocin-resistance (ZeoR) genes into GSH candidates. (**A**) The knock-in scheme for AcGFP1 and ZeoR expression units is shown; the donor vectors harboring AcGFP1 and ZeoR genes under control of the 121 promoter were transfected into Pv11 cells. (**B**) The survival rate after desiccation–rehydration treatment is shown for each knock-in cell line. (**C**) The proliferation rate of the knock-in cell lines is shown. Values are expressed as mean ± SD; n = 4 in each group. N.S., not significant.

**Figure 4 genes-13-00406-f004:**
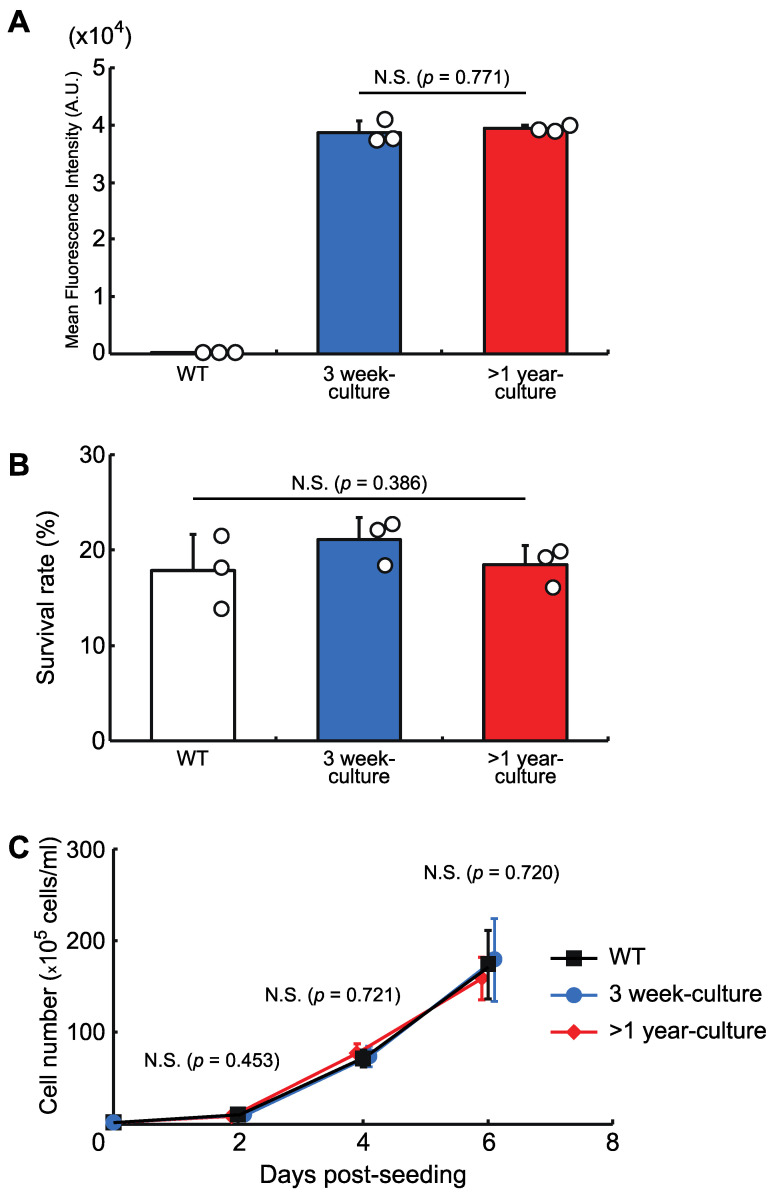
The effects of long-term culture after knock-in at the GSH, Chr1:21164645. (**A**) AcGFP1-expression stability after more than one year in culture without zeocin. (**B**) Cell survival rate following desiccation–rehydration treatment after more than one year in culture without zeocin. (**C**) Cell proliferation rate after more than one year in culture without zeocin. Values are expressed as mean ± SD; n = 3 in each group in (**A**,**B**); n = 4 in each group in (**C**). N.S., not significant.

**Figure 5 genes-13-00406-f005:**
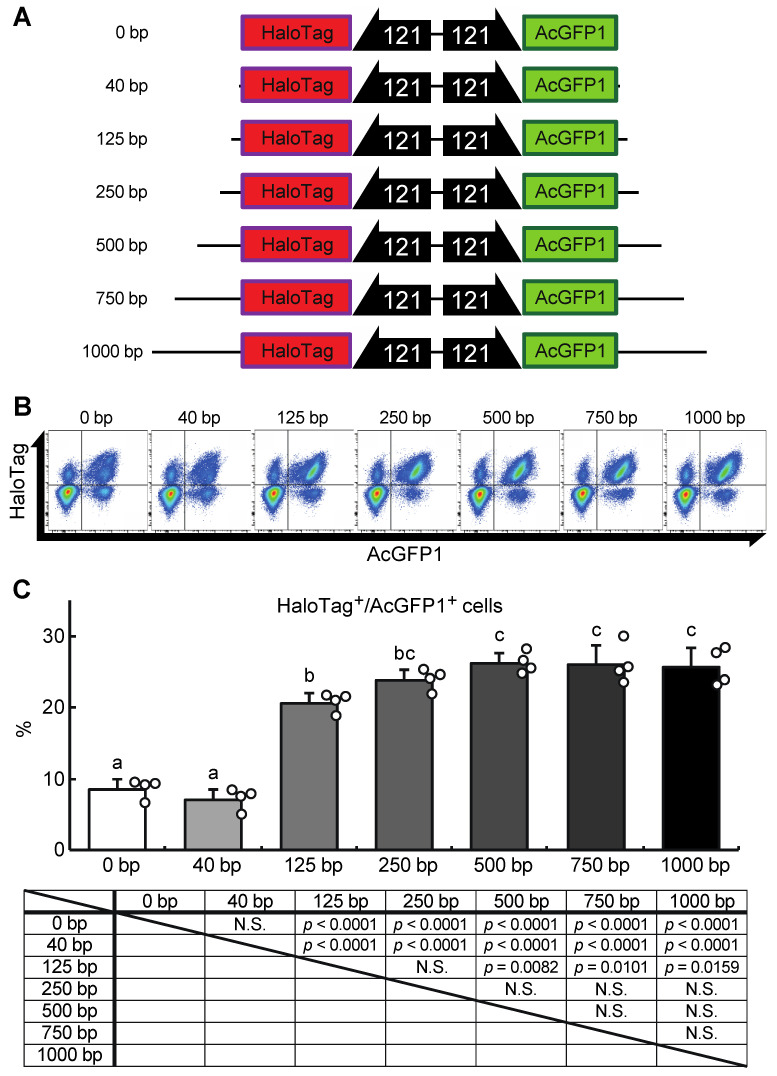
A donor vector construct for GOI expression and optimization of the homology arm length for maximum knock-in efficiency at the Chr1:21164645 site. (**A**) Schematic outline of donor vectors with different homology arm lengths in the range 0–1000 bp. (**B**) Representative dot plot data of transfected cells after zeocin selection showing AcGFP1 and HaloTag fluorescence. (**C**) The proportions of AcGFP1^+^ and HaloTag^+^ cells in the live-cell population analyzed using a flow cytometer after 10 days in culture with zeocin selection, and the result of the statistical analysis. Values are expressed as mean ± SD; n = 4 in each group. N.S., not significant. Different letters above each bar indicate significant differences among groups at *p* = 0.05 as shown in the statistical analysis. The darker shade indicates the longer homology arm length.

## Data Availability

Raw sequencing data of the whole-genome libraries and the ATAC-seq generated in this study are available at NCBI GEO under accession number GSE190481 (token: qvslwuusvrihjyt).

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
