# Peer review of "Identification of Genomic Safe Harbors in the Anhydrobiotic Cell Line, Pv11"

_genes, 2022, doi:10.3390/genes13030406_

Round 1

Reviewer 1 Report

The manuscript by Miyata et al deals with the identification and characterization of novel Genomic Safe Harbors (GSHs) in the dry-preservable cell line Pv11, derived from Polypedilum vanderplanki. GSHs are sites of a genome where integration of exogenous DNA is not detrimental for cell viability, and where the exogenous DNA can be transcriptional active. The authors identify four of such loci, all in chromosome 1, and further characterize more extensively three of them. This work provides a good foundation for a more controlled engineering of this cell line, both for industrial applications and for the understanding of anhydrobiosis.  The following are only minor points that can improve the article in my opinion.

-Figure 4A: Authors show three datapoints around the top of each bar, and it is not clear if these are three independent clones or three measurements of the same clone. I think it would be interesting to know how much clonal variability is there. Authors should also mention whether the long term growth was in the presence of selection (zeocin). 

-Another question that a potential reader may have is about the effect of anhydrosis in the expression of an exogenous DNA integrated at the GSH, which is a relevant point because the most interesting feature of this line is its ability to resist desiccation. Have the authors studied gene expression of foreign genes at the GSHs after repetitive cycles of anydrobiosis?

-something that I have also missed in this study is some further confirmation of the low impact of the integration of exogenous DNA at GSHs. Authors focus on growth profile and survival rate after desiccation, which are sensible parameters, but some investigation on the possible impact on the transcriptional activity of neighboring genes/rest of the genome would be useful. Authors may have some RNA-Seq data that they would like to show, or at least the expression levels of some neighbouring genes.

-In section 2.1. of Materials and Methods, please provide more info about growth conditions (temperature, CO2%, etc) 

-Figure 5 - Please indicate in figure legend which GSH was targeted by these constructs.

-Do authors have any indication that the different GSHs that they describe (alpha, beta, delta and gamma) can be used simultaneously with different constructs? Can the authors comment on the possibility of inserting more than one plasmid copy at the same location?

Typos and text edits

Line 293: it should read knock-in

Line 294: font should be regular, not bold

Figure S2: Text in the figure, Chr1: 280397-Knock-in (correct “knock”)

Line 142: it should read “performed”

Author Response

Point-by-point response

Reviewer 1:

>> -Figure 4A: Authors show three datapoints around the top of each bar, and it is not clear if these are three independent clones or three measurements of the same clone. I think it would be interesting to know how much clonal variability is there. Authors should also mention whether the long term growth was in the presence of selection (zeocin).

In Figures 4A and B, those data points show the results of culturing each clonal cell line in three separate flasks and analyzing them independently. This means that the experiment was performed in three biological replicates. We added the explanation in the legend of Figure 4.

We also added the culture condition of “3 week-culture” and “>1 year-culture” samples in section 2.11.

>> -Another question that a potential reader may have is about the effect of anhydrosis in the expression of an exogenous DNA integrated at the GSH, which is a relevant point because the most interesting feature of this line is its ability to resist desiccation. Have the authors studied gene expression of foreign genes at the GSHs after repetitive cycles of anydrobiosis?

Thank you for an important comment on our ongoing project. Actually, we are making a biotechnological product using transgenic Pv11 cells with exogenous DNAs at Chr1:21164645. We found that rehydrated cells could start to synthesyze the protein coded by the exogenous DNAs, and, within 24 hours, the protein amount seemed to reach the comparative level to that of intact cells. We are now carefully analyzing the expression time-course for further improvement of our Pv11-product. We completely agree with Reviewer 1's suggestion, but would like to avoid including these detailed data in the main text, as it is the content of a patent application that is currently being prepared.

>> -something that I have also missed in this study is some further confirmation of the low impact of the integration of exogenous DNA at GSHs. Authors focus on growth profile and survival rate after desiccation, which are sensible parameters, but some investigation on the possible impact on the transcriptional activity of neighboring genes/rest of the genome would be useful. Authors may have some RNA-Seq data that they would like to show, or at least the expression levels of some neighbouring genes.

Thank you for pointing this out. We agree that insertion of exogenous genes into genomic sites can occasionally changes in genomic structure or gene expression pattern, which in tern can cause alteration in cellular phenotypes (e.g., oncogenesis). Such phenotypic changes are a major concern, especially in human gene therapy applications. In general, global gene expression patterns are assessed to confirm that the likelihood of any potential phenotypic changes in the future is low. In other word, the most important point is whether the intrinsic cellular phenotypes can be maintained after transgenesis.

Intrinsic characteristics of Pv11 cells, namely the ability to anhydrobiosis and proliferation, were maintained in the cells inserted exogenous DNA into the GSH site even after one year (Figs 3 and 4). We believe that the current data is sufficient to derive our conclusion even if the expression pattern of endogenous genes is somewhat altered in the transgenic cell lines. We added the disccussion above into page 12 (line 437-445).

>> -In section 2.1. of Materials and Methods, please provide more info about growth conditions (temperature, CO2%, etc)

We added the detailed culture conditions in section 2.1.

>> -Figure 5 - Please indicate in figure legend which GSH was targeted by these constructs.

We added the expression “at the Chr1:21164645” in the legend of Figure 5.

>> -Do authors have any indication that the different GSHs that they describe (alpha, beta, delta and gamma) can be used simultaneously with different constructs? Can the authors comment on the possibility of inserting more than one plasmid copy at the same location?

I appreciate your comment. Actually, we are trying to establish a double knock-in cell line at the Chr1:21155382 and Chr1:21164645 sites by inserting GOIs independently instead of simaltaneously. As shown in Figure 5, the efficiency of precise integration was not so high even when the homology arm length was much longer than the original PITCh technique. So, we decided to insert exogenous DNA materials into the two sites one by one; first, establishment of a Chr1:21164645-knock-in cell line, and then, insertion at the Chr1:21155382 site of the parental Chr1:21164645-knock-in line. Then, we will analyze the protein expression levels between the single and double knock-in cell lines. Based on the preliminary data, we added the discussion on the possibility of double knock-in or multi-copy integration (page 11-12, line 402-446).

Reviewer 2 Report

The authors worked with a cell line, Pv11, from a desiccation-resistant dipteran species, of which both the larvae and, interestingly, also the cell line are resistant to desiccation. This topic is very interesting because all the proteins that are present in these cells can be stored at room temperature and remain preserved when the cells containing them are desiccated. In this paper, the authors isolated cell clones from a cell pool containing random transgenic insertions (the KH cell pool). They found two clones, B2 and 4C that were less impaired in their growth phenotype that the rest of the cell pool members were, thus identifying clones that may have transgenic insertions in Genomic Safe Harbor positions. They identified the exact landing sites by sequencing and later targeted these sites to test if the insertion of genes leaves the cellular growth phenotype intact (and hence proving that the sites are indeed Genomic Save Harbors). The authors found that several of these sites can indeed be used to insert foreign genes without impacting the physiological properties (cell growth and desiccation resistance) of these cells. The importance of this work is that the identification of safe landing sites for genes can be used to build protein factories, and the resulting proteins can be stored at room temperature, as long as they remain in the dried cells.

Critique:

The paper is very clearly and comprehensively written, and the data look very good and support the conclusions made in the paper.

I found a few small issues, which I wrote in the PDF that I attach. I want to mention a few here in writing:

1) Line 86: Perhaps define AcGFP1. I know what GFP is, and I guessed that AcGFP1 is probably GFP as well.

2) Figure 1 A: I would write KH cells (with an s at the end)

3) Figure 2 C: Why did you call the sites "a b g d" and not "a b c d"? I know that Cyrillic does not have a "c" and that "g" indeed comes before "d". I don't mind it, but I thought it looked curious.

4) Line 299: Something happened at the end of the sentence. Maybe the "/" should be replaced with ")." 

5) In Figure 5C, please explain in the figure legend what the shading of the bars means and what the letters and combinations thereof mean (a, b, bc, c).

Otherwise, this was a very good read, and the science is solid. Good work!

Author Response

Point-by-point response

Reviewer 2:

>> 1) Line 86: Perhaps define AcGFP1. I know what GFP is, and I guessed that AcGFP1 is probably GFP as well.

We added the definition of AcGFP1; Aequorea coerulescens GFP. AcGFP1 is a variety of GFP. Originally, GFP was isolated from a jellyfish, Aequorea victoria, but AcGFP1 was isolated from a different species of jellyfish, Aequorea coerulescens.

>> 2) Figure 1 A: I would write KH cells (with an s at the end)

 We corrected the point.

>> 3) Figure 2 C: Why did you call the sites "a b g d" and not "a b c d"? I know that Cyrillic does not have a "c" and that "g" indeed comes before "d". I don't mind it, but I thought it looked curious.

Actually, our original manuscript showed “alpha, beta, delta and gamma” in Figure 2 as shown below, and Reviewer 1 also recognized the symbol correctly. So, we think that the symbols were displayed wrongly on Reviewer 2’s viewer. Could you please check the original manuscript with an appropriate viewer?

>> 4) Line 299: Something happened at the end of the sentence. Maybe the "/" should be replaced with ")."

We corrected the wrong display.

>> 5) In Figure 5C, please explain in the figure legend what the shading of the bars means and what the letters and combinations thereof mean (a, b, bc, c).

We are sorry for our insufficient explanation in Figure 5C. We added the explanations in the legend. Further, we noticed that ‘Statistical Analysis’ section was absent from Materials and Methods section. So, we added the section as ‘2.13. Statistical Analysis.’
